# Deep learning approach based on superpixel segmentation assisted labeling for automatic pressure ulcer diagnosis

**Che Wei Chang**[1,2]*, **Mesakh Christian**[3], **Dun Hao Chang**[2,4], **Feipei Lai**[1,3], **Tom J. Liu**[1,5], **Yo Shen Chen**[2], **Wei Jen Chen**[1]

**1** Graduate Institute of Biomedical Electronics & Bioinformatics, National Taiwan University, Taipei, Taiwan, **2** Division of Plastic and Reconstructive Surgery, Department of Surgery, Far Eastern Memorial Hospital, New Taipei, Taiwan, **3** Department of Computer Science & Information Engineering, National Taiwan University, Taipei, Taiwan, **4** Department of Information Management, Yuan Ze University, Taoyuan City, Taiwan, **5** Division of Plastic Surgery, Department of Surgery, Fu Jen Catholic University Hospital, Fu Jen Catholic University, New Taipei City, Taiwan

* b92401068@gmail.com

**Data Availability Statement:** Data cannot be shared publicly because of regulation of institutional review board of Far Eastern Memorial Hospital. Data are available from the institutional

## Abstract

A pressure ulcer is an injury of the skin and underlying tissues adjacent to a bony eminence. Patients who suffer from this disease may have difficulty accessing medical care. Recently, the COVID-19 pandemic has exacerbated this situation. Automatic diagnosis based on machine learning (ML) brings promising solutions. Traditional ML requires complicated pre-processing steps for feature extraction. Its clinical applications are thus limited to particular datasets. Deep learning (DL), which extracts features from convolution layers, can embrace larger datasets that might be deliberately excluded in traditional algorithms. However, DL requires large sets of domain specific labeled data for training. Labeling various tissues of pressure ulcers is a challenge even for experienced plastic surgeons. We propose a super-pixel-assisted, region-based method of labeling images for tissue classification. The boundary-based method is applied to create a dataset for wound and re-epithelialization (re-ep) segmentation. Five popular DL models (U-Net, DeeplabV3, PsPNet, FPN, and Mask R-CNN) with encoder (ResNet-101) were trained on the two datasets. A total of 2836 images of pressure ulcers were labeled for tissue classification, while 2893 images were labeled for wound and re-ep segmentation. All five models had satisfactory results. DeeplabV3 had the best performance on both tasks with a precision of 0.9915, recall of 0.9915 and accuracy of 0.9957 on the tissue classification; and a precision of 0.9888, recall of 0.9887 and accuracy of 0.9925 on the wound and re-ep segmentation task. Combining segmentation results with clinical data, our algorithm can detect the signs of wound healing, monitor the progress of healing, estimate the wound size, and suggest the need for surgical debridement.

## Introduction

A pressure ulcer or pressure sore is a pressure-induced injury of the skin or underlying tissue adjacent to bony eminences. These injuries appear most commonly on the sacral area, followed

review board (contact via phone: +886-2-8966-7000 ext. 2152 and the e-mail: irb@mail.femh.org.tw.) for researchers who meet the criteria for access to confidential data. Alternatively, we uploaded the images labeled by our proposed methods (boundary-based and region-based) to the data repository site. The DOIs by two different methods are S1 Dataset. Region-based label method https://doi.org/10.6084/m9.figshare.17206904.v1 and S2 Dataset. Boundary-based label method https://doi.org/10.6084/m9.figshare.17206940.v1. The images are collected from the open dataset of wound, the Medetec. Those labeled images were part of our testing sets. Researchers can train DL models with these two datasets to replicate our study results.

**Funding:** This work was supported by the Innovation Project of Far Eastern Memorial Hospital (Grant No. PI20200002). The funders had no role in study design, data collection and analysis, decision to publish, or preparation of the manuscript.

**Competing interests:** The authors have declared that no competing interests exist.

by the heel and occipital area. The causal mechanism is application of pressure to the skin exceeding end capillary pressure, which is 20–30 mmHg. Since no blood can flow in, the tissues gradually undergo necrosis. However, simply applying that level of pressure to the skin may not necessarily result in a pressure ulcer. It was found that pressure must persist for more than 2 hours to cause irreversible ischemic damage [1]. Etiological factors in the development of pressure ulcers include medical conditions, such as stroke, spinal cord injury, shock, malnutrition, antidepressant use, and catheter use [2]. Major surgery or trauma are also contributing factors.

In the United States, pressure ulcers affect around 3 million adults annually [3]. In Europe, the prevalence rate in hospitals ranges from 8.3% to 23% across different countries [4]. The prevalence rate is higher in long-term care facilities than in hospitals [2, 5]. Although this tissue had got attention from the public, the prevalence of pressure ulcers has remained unchanged over the last decade [3, 6]. As the life span of humans increases, the affected population is expected to rise. Another problem of growing concern is that patients with pressure ulcers often have difficulty accessing medical care because of their underlying diseases. This condition has been exacerbated by the Covid-19 pandemic.

Machine learning (ML) has many applications in the field of medicine, such as in drug development and disease diagnosis [7–11]. Automatic wound diagnosis based on ML can provide an objective assessment of wound status, such as wound size and healing stage, thereby easing the load of medical providers and providing timely suggestions for patients when medical care is not easily accessed.

## Related work

There are two major tasks for which ML can be beneficially employed in the treatment of pressure ulcers. The first is wound segmentation; the other is tissue classification [12–14]. Wound segmentation is necessary to separate the wound area from the non-wound area. There are three standard methods to achieve this result: threshold, edge-based and region-based segmentation. Threshold segmentation applies algorithms to separate pixels of the wound in a wound image from pixels of normal skin, which are more homogeneous in intensity [15–17]. Edge-based segmentation initializes an approximate shape as a border, such as an ellipse or circle. At each iteration, the border is adjusted to minimize the energy function so that the final border fits the edge of the wound [18, 19]. Region-based segmentation selects a small region of the wound and gradually recruits adjacent pixels with similar color energy [20, 21]. The above methods can be further combined to acquire better results [22, 23]. After successful wound segmentation, the size of a wound can be further calculated. Although the size of a wound is the most direct information about wound healing, it is of less clinical usage than the tissue composition of a wound. A pressure ulcer usually takes weeks to months to decrease in size.

Tissue classification, as another assessment of wound healing, is the second important goal for ML models. Before performing tissue classification, images of wounds are usually converted from RGB to other color spaces. These three primary colors (RGB) are correlated to each other. The most common formats of color spaces are HSV, followed by CIELAB and YCbCr [24]. Usually, the luminance components such as the Y channel or L channel are excluded to decrease the light effect in various clinical situations. The next step is to calculate the probability map of pixels of different tissues. ML methods, such as k-means, support vector machine, random forest or Bayesian are used to learn the threshold and classify tissues.

In the literature simultaneously addressing both wound segmentation and tissue classification, although the two tasks seem to be similar, some authors have used different preprocessing methods to get better results. For example, García-Zapirain et al. converted the images to the HSI color space for wound segmentation and used linear combinations of discrete Gaussians

for tissue classification [25]. Elmogy et al. suggested YCbCr images for wound segmentation, while recommending RGB images for tissue classification [26]. Veredas et al. used color histograms for wound segmentation and Bayes rule for tissue classification [16]. Rajathi et al. used deep learning for tissue classification but a simple gradient descent for wound segmentation [27]. These studies demonstrate important skills in data preprocessing and features extraction. The problem in these approaches is, even with good results, they can only be applied to their own datasets. This diminishes their clinical application. Secondly, most studies of ML in wound diagnosis have considered chronic wounds in general rather than those focusing specifically on pressure ulcers. Little evidence exists showing that models trained from ulcers of mixed etiology can equally be used for pressure ulcers.

Although the structures of deep learning (DL) models are capable of handling large quantities of data with many channels, such datasets are often trimmed down for ML to get better results. For example, the lumination channels of color spaces (L channel of CIELab and Y channel of YCbCr) are usually excluded for ML models to decrease the effects under different light conditions. For DL models, instead, lumination channels are input with other channels in order to comprehend various light conditions. DL models, on the other hand, require a large amount of labeled data to achieve good results and prevent overfitting. How to systematically label the data then becomes a very critical issue.

Very few articles have addressed the details of labeling images, only describing them as "labeled by medical experts." In our study, we reveal our step-by-step approach of data labeling and the revisions we made during labeling. This labeling method is suitable for any images of pressure ulcers, not just for our datasets. In the second part, we input more than 2,800 images of pressure ulcers into some of the most popular and influential DL models. We combined the results from different models to create an automatic pressure ulcer diagnosis system.

## Methods

### Image acquisition

This study was approved by the research ethics review committee of Far Eastern Hospital (No. 109145-E). We searched the electric medical records of patients from January of 2016 through December of 2020, with ICD9 codes from 707.00 to 707.10. These codes identify a pressure ulcer and its location; for instance, 707.03 indicates ulcer, pressure, sacrum, i.e., a pressure sore located at the sacrum. We collected the images of pressure ulcers from electronic medical record of progress notes, operation notes, discharge notes, and emergency department charts. All images were 3600*2700 pixels and saved as jpg files. The images were taken by various types of devices, and the conditions of taking pictures including illumination, background, distance, and angle were also diverse. These images were assigned random numbers for de-identification and presented to plastic surgeons for labeling.

An image was initially excluded if the wound was dressed with ointment, covered with dressing, actively bleeding, obscured by hematoma; the wound presented with obvious pus; the image was out of focus or taken under poor light conditions.

### Boundary-based labeling

Our initial method of labeling a pressure ulcer is similar to pizza-making (Fig 1). The first step is to make the round base of dough and the second step is to sequentially add the toppings, such as sausage, mushroom, and olive. So akin to making a pizza, the border of the whole pressure ulcer was first drawn as ulceration. Then the boundaries of the different tissues, such as granulation, eschar, slough inside ulceration, were labeled in order. A pressure ulcer can be described by the different tissue types constituting it, such as by the following relation:

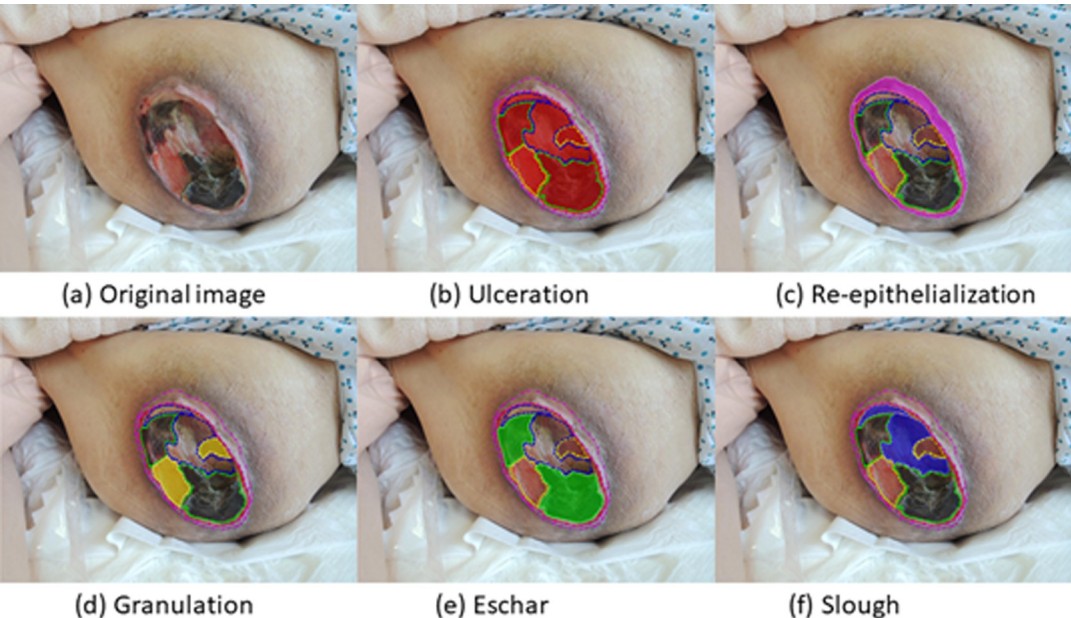

**Fig 1. Pizza-making labeling method.** The border of the ulceration is first drawn, followed by borders of the different tissues inside the ulceration. Re-ep (c) is drawn where the wound image indicates its existence.

Ulceration = (Granulation ∩ Eschar ∩ Slough ∩ Undefined). A pressure ulcer may not have all types of tissues (Table 1).

Peri-wound tissue, re-epithelialization, was labeled. Re-epithelialization (re-ep) is a term to describe the resurfacing of a wound with new epithelium [28]. It is the essential sign of wound healing starts 24 hours after injury and continues until the healing process covers the entire wound surface. The labeling was done using *Labelme* in Python 3.6 with a mouse or pen tablet. All images were co-labeled by two of four plastic surgeons who yielded a single, agreed upon label. Any image was excluded when the two surgeons could not reach consensus on the label.

## Evaluation metrics

Before describing our method of data preprocessing, we need to define the evaluation metrics we used. Five metrics were employed: Dice's coefficient (F1 score), intersection over union

**Table 1. Tissues of pressure ulcer inside and peri-wound.**

| Type | Color | Description |
|---|---|---|
| Granulation (Inside) | Beefy red to pink | 1. Healthy tissue in wounds |
| | | 2. Collagen, fibroblast, capillary |
| | | 3. Indication of wound healing |
| Eschar (Inside) | Black to brown | 1. Desiccate, nonviable tissues from skin, fat, tendon, or muscle |
| | | 2. Usually clear margin if stable |
| Slough (Inside) | Mostly yellow, could be white, tan, grey or green | 1. Most varied appearance |
| | | 2. Partially dead or complete necrotic |
| | | 3. Mix with fibrin, biofilm or bacteria |
| Re-epithelialization (Peri-wound) | Light pink to purple | 1. Formation of wound bed matrix |
| | | 2. Migration of keratinocytes |
| | | 3. Indication of wound healing |

(IoU), precision, recall and accuracy. Dice's coefficient (DC) and IoU are two common metrics to assess segmentation performance, whereas precision, recall and accuracy are the metrics of assessing classification results. These metrics are defined using true positive (TP), false positive (FP), true negative (TN) and false negative (FN) predictions for any input image considered, as follows:

**DC (F1 score)** is twice the area of the intersection of the ground truth and prediction divided by the sum of their areas. It is given by:

$$DC = \frac{2|Area(Predict) \cap Area(Ground\ truth)|}{|Area(predict)| + |Area(Ground\ truth)|} \ or \ \frac{2TP}{2TP + FP + FN}.$$

The **intersection over union (IoU)** denotes the area of the intersection of the ground truth and prediction divided by the area of their union. It is given by:

$$IoU = \frac{|Area(Predict) \cap Area(Ground\ truth)|}{|Area(predict) \cup Area(Ground\ truth)|} \ or \ \frac{TP}{TP + FP + FN}.$$

**Precision** is defined as the ratio of the actual wound pixels that the model correctly classified to all predicted wound pixels. It is also called the **positive predicted value** (PPV) and given by:

$$Precision = \frac{TP}{TP + FP}.$$

**Recall** is defined as the ratio of the actual wound pixels that are correctly classified to all actual wound pixels. It is also called **sensitivity** and given by:

$$Recall = \frac{TP}{TP + FN}.$$

Accuracy denotes the percentage of correctly classified pixels. It is given by:

$$Accuracy = \frac{TP + TN}{TP + FP + TN + FN}.$$

## Different combinations of classes

The above labeling methods created a detailed annotation with different types of tissues inside and outside of the ulceration. However, the informative label data generated a challenge of repeat annotation on the same pixels. Different classes needed to be trained together, and at least two modes were required. There were multiple possible combinations of different classes into two groups. To decide which combination of groups to select not only depends on final segmentation results, but which tissue is more important for diagnosis. Granulation and re-ep are the two most important tissues for detecting wound healing. However, re-ep had fewer total numbers of pixels than other tissues (S1 Fig) and, the shape of re-ep is usually a ring or crescent, making it difficult to predict accurately.

We tried two practical class combinations to test which models had better segmentation performance for re-ep. Re-ep was trained with ulceration in model 1 whereas re-ep was trained with granulation, slough and eschar in model 2. The test dataset contained 755 labeled images of pressure ulcers and was split at a ratio of 7:2:1 for training, validation and testing. The data was input into U-Net with ResNet101 for training.

In S2 and S3 Figs, they showed that model 1 (recall: 0.6291, DC: 0.7716) had better sensitivity and DC for re-ep than model 2 (recall: 0.5685, DC: 0.7244). Based on this result, re-ep and

ulceration were trained in one model, and tissues inside ulceration were trained in the other model.

## Region-based labeling

Pizza-making is a straight-forward, boundary-based labeling method. Nevertheless, we found it challenging for plastic surgeons to acquire consensus when they labeled tissues (granulation, slough, eschar) inside the ulceration. The composition of tissues on the wound bed is a subtle transition. It is sometimes hard to draw boundaries between different tissues. It was also difficult for two plastic surgeons to reach consensus on these borders.

We introduced superpixel segmentation to define meaningful regions among tissues. We tried to reduce inconsistency of the borders drawn by different labelers. There are different algorithms to achieve this goal. Simple linear iterative clustering (SLIC) is the most popular method [29]. It groups pixels in a five dimensional space (l,a,b,x,y), where (l,a,b) are the color vectors in the CIELAB color space and (x,y) are the pixel positions. The desired number of equally-sized superpixels K are defined, and the grid interval is S. Compactness (m) is a variable that controls the weight of the spatial term. The center of K superpixel $C_k$ is $(l_k,a_k,b_k,x_k,y_k)$. Each pixel $I_i$ is be represented as $(l_i,a_i,b_i,x_i,y_i)$. The Euclidean distances D are defined as follows:

$$D_{color} = \sqrt{(l_i - l_k)^2 + (a_i - a_k)^2 + (b_i - b_k)^2}$$

$$D_{xy} = \sqrt{(x_i - x_k)^2 + (y_i - y_k)^2}$$

$$D = D_{color} + \frac{m}{S} D_{xy}$$

The process of associating pixels with the nearest cluster center computed the boundary repeatedly until converge.

To test whether the dataset preprocessed by SLIC had a positive effect on the models to segment different tissues (granulation, slough and eschar), the 755 images previously labeled with *labelme* were used. All images were reduced in size to 1000*750 pixels but filled with black background to 1000*1000 pixels for model training. The smaller images can be processed with SLIC in laptops without powerful GPUs. Images were processed by SLIC with K ranging from 800 to 1000, and m = 10, to form superpixels (regions). Fig 2 shows an illustrative example of results with different settings of the number of superpixels (K). After preprocessing with SLIC, images were presented to the same two plastic surgeons who sorted the regions into their classes.

The dataset labeled with *labelme* and the dataset labeled with SLIC preprocessing were both input into U-Net with the same fine-tune parameters to yield model 3 and model 4.

The performance of tissue classification of the two models is shown in Table 2. The model 4 trained on the reduced images with SLIC preprocessing had better results on all evaluation metrics. The results do not necessarily mean that model 4 has better segmentation performance in real-wound images since the ground truth of the two datasets are different. We used 75 new testing images of pressure ulcers to make a direct comparison. model 4 still had more satisfactory segmentation results and predicted fewer undefined pixels than model 3, as illustrated in Fig 3.

Based on the above results, we revised the boundary-based labeling for all classes (pizza-making method) to boundary-based labeling for re-ep and ulceration; region-based labeling

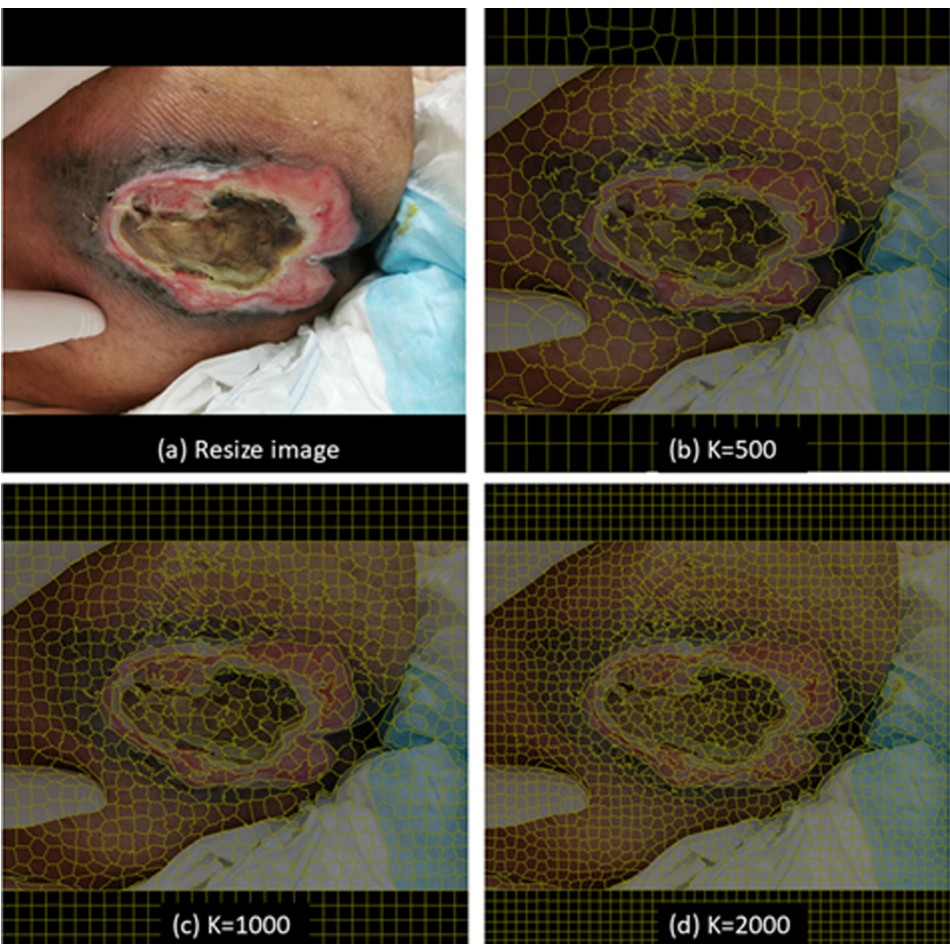

**Fig 2. Different number of superpixels (K).** (a) Resize image. (b) K = 500. (c) K = 1000. (d) K = 2000.

(SLIC preprocessing) for granulation, slough and eschar. One dataset (boundary-based) was used to train models for wound segmentation. The other dataset (region-based) was applied to trained models for tissue classification (Fig 4). Images of the boundary-based dataset are their original sizes, and the images of the region-based dataset are reduced in size as described above. The two datasets were also split at a ratio of 7:2:1 for training, validation and testing. Training and validation images were used to execute three-fold cross validation and the testing set was preserved.

**Table 2. Tissue classification from models with different labeling methods.**

|  | Model 3 (*Labelme*) | Model 4 (SLIC) |
|---|---|---|
| IoU score | 0.3921 | 0.4635 |
| F1 score | 0.4336 | 0.5129 |
| Precision | 0.5036 | 0.5622 |
| Recall | 0.7979 | 0.8233 |
| Accuracy | 0.9897 | 0.9927 |
| Loss | 0.5664 | 0.4870 |

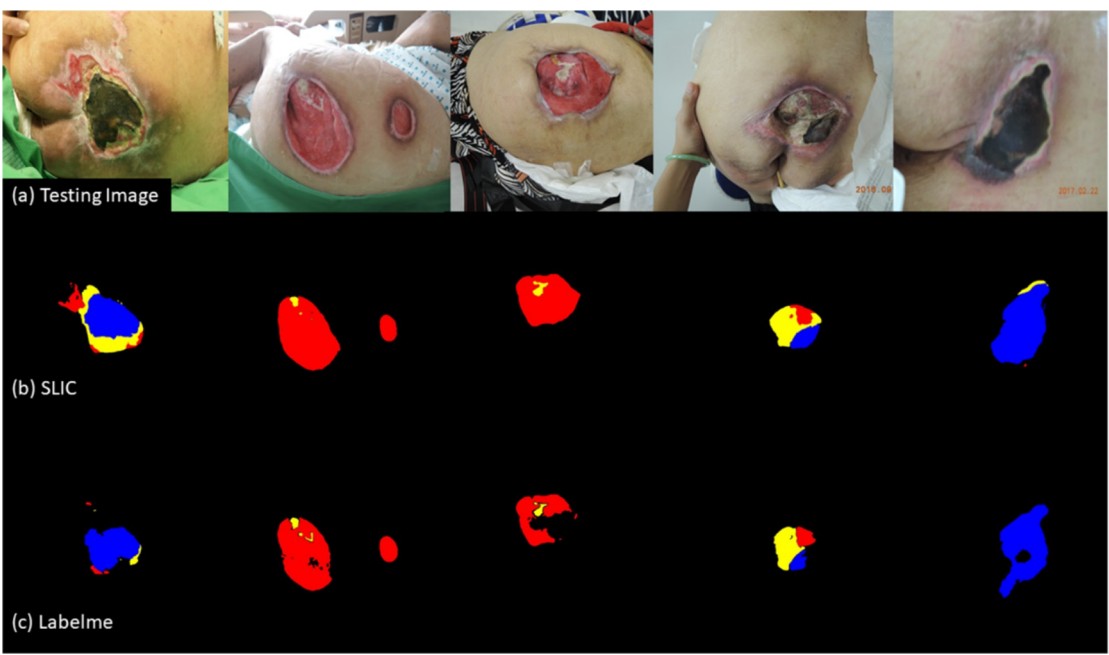

**Fig 3. Tissue classification performance on testing images of U-Net with different labeling methods.** (a) Original images of pressure ulcers. (b) Classifications of Model 4 (SLIC). (c) Classifications of Model 3 (*Labelme*). Granulation: red; Slough: yellow; Eschar: blue.

## Deep learning models

The two datasets were input into five well-developed DL models to compare their performance on the two tasks. All models were combined with ResNet101 as their encoder. ResNet101 is a powerful encoder that can be compatible with any deep learning models, and easy to optimize. We also initialized them with pre-trained model weights derived from large-scale object detection, segmentation and captioning datasets, such as ImageNet and COCO. The standard image augmentations of images we used were rotating, shifting, scaling, gaussian blurring, and contrast normalization. All the models were trained in the Taiwan Computing Cloud, using their container service, on a server configured with four *NVIDIA TESLA V100* GPU cards, a 16 core *Intel XEON Gold 61* CPU and 360 GB RAM.

**U-Net.** U-Net proposed by Ronneberger et al. is the most popular semantic segmentation model in the medical field [30]. Its architecture, which involves a series of encodings followed by a decoding process, resembles the letter U. Many models have been derived from U-Net structures, such as FCNet, SegNet, DeconNet, V-Net, U-Net++. The standard Dice loss was chosen as the loss function. The formula is given by:

$$L(TP, FP, FN) = 1 - \frac{2TP + \epsilon}{2TP + FP + FN + \epsilon}.$$

wThe $\epsilon$ term is used to avoid the issue of dividing by 0 when precision and recall are empty.

**DeeplabV3.** DeeplabV3 was proposed by Chen et al [31]. Multiple convolution layers are a structure designed to capture high dimensional features, but they lose details when layers go deeper. Atrous convolution is a solution created to solve this problem. Deeplab V3 is composed of multiple scales of atrous convolutions (spatial pyramid pooling) to preserve details

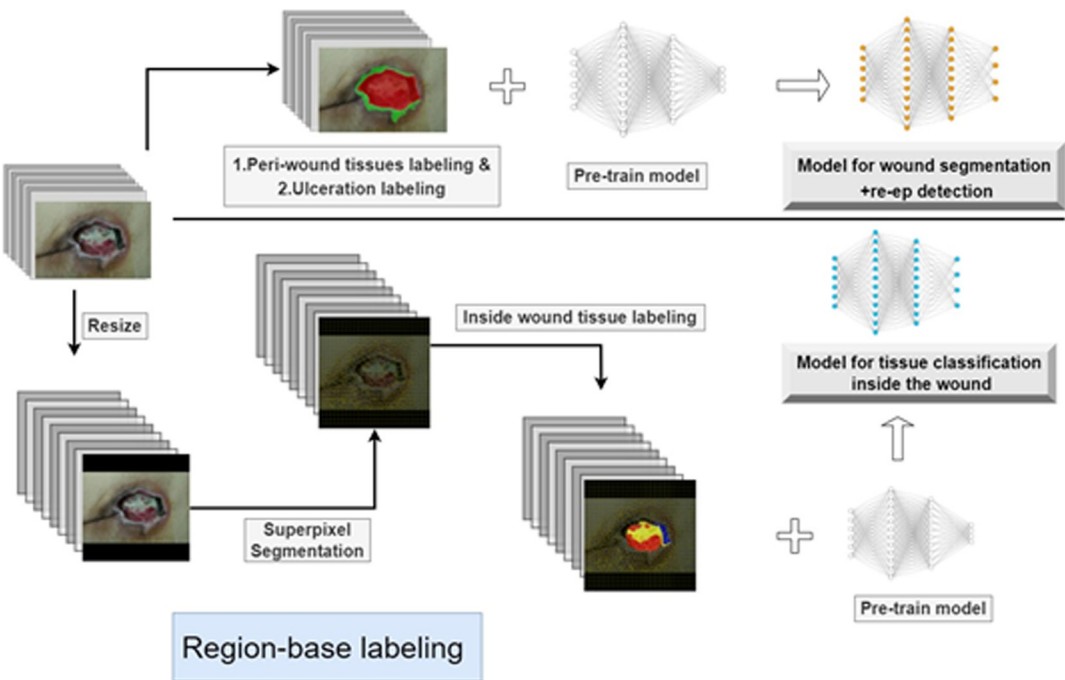

**Fig 4. The workflow of labeling and training.** Boundary-based labeling was used for peri-wound tissue and ulceration. Region-based labeling was applied for inside wound tissues.

and acquire high dimensional features. The standard Dice loss was chosen as the loss function of U-Net.

**Pyramid scene parsing (PsPnet).** PsPNet was proposed by Zhao et al [32]. After passing through CNN layers, such as ResNet pretrained with a dilated network for feature extraction, the model adopts whole and multiple sub-regional features from larger size to smaller size, which calls a pyramid pooling module. Then, the model up-samples all the sub-regional features and then concatenates the original feature map to form a single layer. The standard Dice loss was chosen as the loss function of U-Net.

**Feature Pyramid Network (FPN).** FPN, proposed by Lin et al., has a modified convolution pathway [33]. It not only has the high level of features of convolutional layers but also combines the features from different levels of up-sampling layers. It works faster than a featurized image pyramid, which has features from different scales of images. It acquires more detail than a single feature map. FPN is a hybrid and modified model of a featurized image pyramid and a single feature map. The standard Dice loss was chosen as the loss function of U-Net.

**Mask R-CNN.** Mask R-CNN was proposed by researchers at Facebook [34]. It is an advanced model of Faster R-CNN and a tool of instance segmentation. Mask R-CNN uses a multi-task loss function given by $L = L_{class} + L_{box} + L_{mask}$. The $L_{class}$ component contains the RPN class loss, a penalty for failure of the Region Proposal Network to separate object prediction from background. The $L_{box}$ measures failure of object localization or bounding by RPN. The last component $L_{mask}$ constitutes the loss from the failure of Mask R-CNN to correctly predict the object mask.

**Table 3. Performance of the five models on wound segmentation and re-ep detection task.**

|            | IoU score | F1 score | Precision | Recall | Accuracy | Loss   |
|------------|-----------|----------|-----------|--------|----------|--------|
| U-Net      | 0.9745    | 0.9867   | 0.9868    | 0.9867 | 0.9911   | 0.0132 |
| DeepLabV3  | 0.9782    | 0.9887   | 0.9888    | 0.9887 | 0.9925   | 0.0112 |
| PsPnet     | 0.9211    | 0.9404   | 0.9317    | 0.9494 | 0.9780   | 0.0595 |
| FPN        | 0.8196    | 0.8939   | 0.8556    | 0.9492 | 0.9346   | 0.1061 |
| Mask R-CNN | 0.8597    | 0.8456   | 0.8345    | 0.8542 | 0.8533   | 0.3742 |

## Results

### Wound segmentation & tissue classification

The first dataset contained 2893 images labeled with the boundary-based method and was used for wound segmentation and re-ep detection. The second dataset included 2836 images labeled with SLIC preprocessing, the region-based method, and used for tissue classification inside the ulceration. The first dataset had more labeled images than the second, because it is challenging to reach consensus when labeling tissues.

The performance of the five models on the wound segmentation and re-ep detection tasks is shown in Table 3. Although the five models all achieved satisfactory results on this task, DeeplabV3 outperformed the other algorithms. As shown in Table 4, DeeplabV3 also did the best on the tissue classification task. In addition to our testing dataset, images from the Medetec Wound Database were used to test the performance on wound segmentation, re-ep detection (Fig 5) and tissue classification inside wounds (S4 Fig). Finally, we chose to use DeeplabV3 for the two main tasks in our automatic diagnosis algorithm (S5 Fig).

However, it is worth noting that although Mask R-CNN is a powerful model, it performed worse than all other models on the tissue classification task. Mask R-CNN outputs instance segmentation by first defining the bounding box for objects and then proposing segmentation masks. If the bounding box is not defined correctly, the mask will not be segmented correctly. In many ulcerations, granulation tissue encases or intermingles with the slough tissue or vice versa, which presents the wrong range of bounding boxes as well as masks, as shown in Fig 6. The actual wound pixels are then underestimated, and recall is affected the most. This will not cause much of an issue for models that perform semantic segmentation.

### Automatic diagnosis

Combining segmentation results and clinical data, the set of algorithms in our automatic diagnostic system provides output concerning four aspects of wound status useful for making treatment decisions: the detection of re-ep signals the initiation of wound healing; the ratio of granulation to ulceration area and the trend of the ratio indicate the stage of wound healing, the estimated size of the wound, and whether surgical debridement is required (S5 Fig). The detection of re-ep relies on the wound segmentation and the segmentation of the area of re-ep

**Table 4. Performance of the five models on tissue classification inside ulcerations.**

|            | IoU score | F1 score | Precision | Recall | Accuracy | Loss   |
|------------|-----------|----------|-----------|--------|----------|--------|
| U-Net      | 0.9830    | 0.9905   | 0.9913    | 0.9897 | 0.9899   | 0.0094 |
| DeepLabV3  | 0.9834    | 0.9915   | 0.9915    | 0.9915 | 0.9957   | 0.0084 |
| PsPnet     | 0.9390    | 0.9753   | 0.9614    | 0.9897 | 0.9899   | 0.0094 |
| FPN        | 0.8252    | 0.9014   | 0.8615    | 0.9507 | 0.9508   | 0.0986 |
| Mask R-CNN | 0.7886    | 0.8480   | 0.9191    | 0.7871 | 0.8903   | 0.2032 |

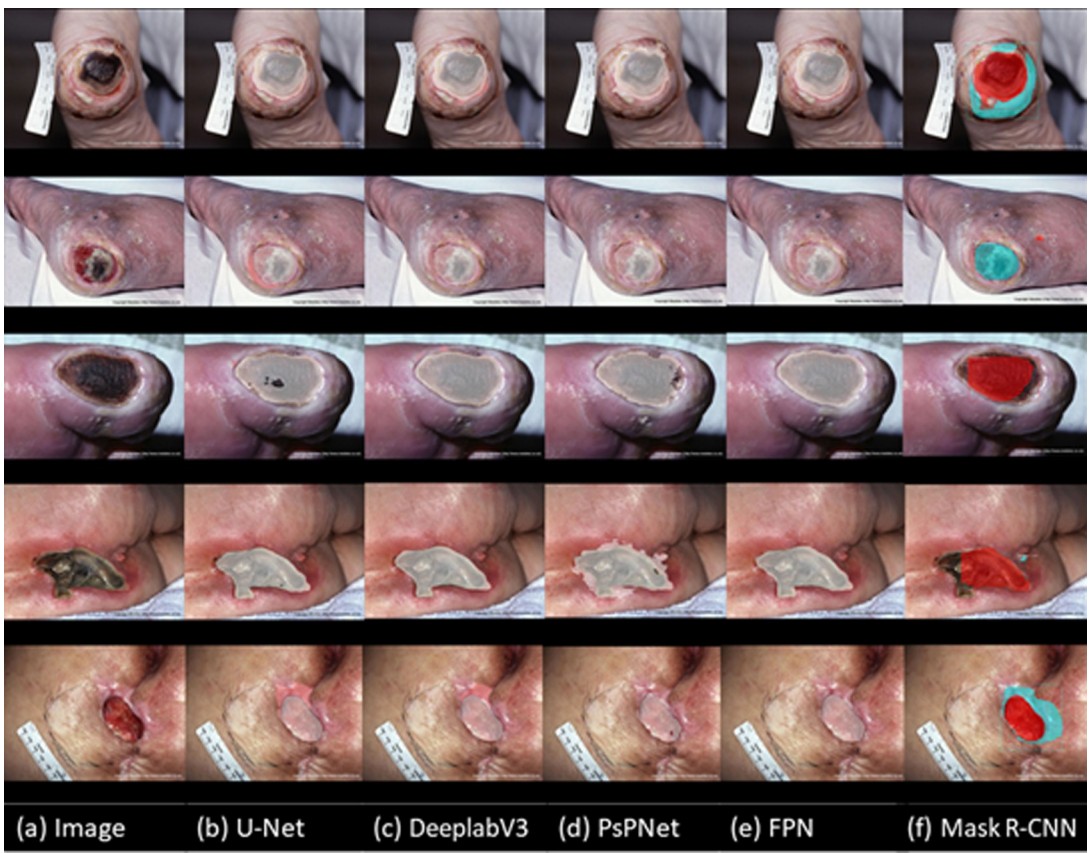

**Fig 5. Collage of results of wound segmentation and re-ep detection from the Medetec Wound Database.**

adjacent to ulceration, both executed using DeeplabV3. Calculating the ratio of granulation to ulceration area requires the results from the wound segmentation and tissue classification performed using DeeplabV3. The coverage rate of granulation tissue provides the strength of wound contraction and indicates the wound is healing. Estimation of the wound size combines the wound segmentation results with parameters extracted from the input images. Assuming that the image was taken at distance D and the projection area of the whole image is A, which can be calculated by:

$$A = Length * width = Length^2 * ratio$$

$$Length = D * Sensor\ size/Focal\ distance$$

If the distance is Dx, the pixel of image is $P_{image}$ and the wound segmentation is $P_{wound}$, the projection of area of the whole image $A_x$ and the wound $A_{wound}$ could be calculated as:

$$A_x = A^{Dx/D},\ A_{wound} = \left(P_{wound}/P_{image}\right) * A_x$$

As the fourth output, whether the pressure ulcer requires surgical debridement depends on the presentation of infection and necrotic tissues [35]. The decision tree consists of three checkboxes in Fig 7. The necrotic tissue can be assessed from the segmentation results (blue box). The infection (gray box) and potential infection of necrotic tissue (orange box) need the input of clinical data with regression and other wound conditions. We deployed all algorithms

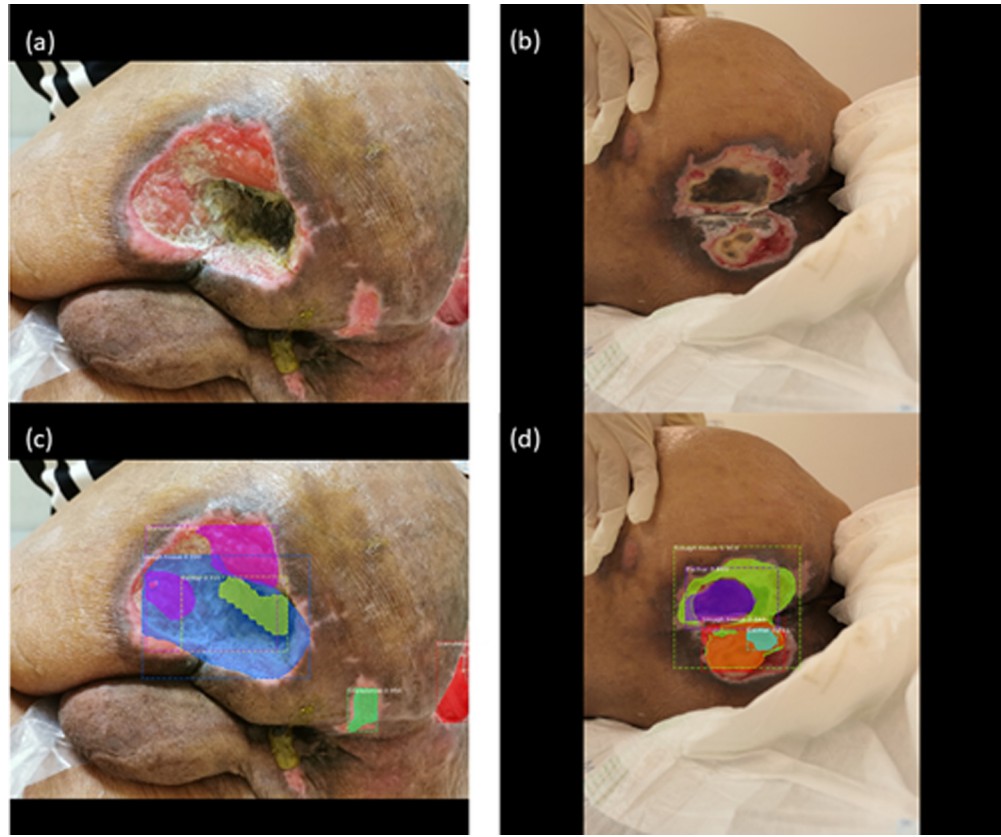

**Fig 6. The wrong range of bounding boxes as well as masks.** (a) The granulation tissues intermingled with slough and eschar. (b) Some eschar was encased by slough and some slough was encased by granulation. (c) and (d) showed poor segmentation results due to the above phenomenon.

and models in the Taiwan Computing Cloud, a virtual computing service, on a dedicated server configured with 4 *Inter XEON GOLD* CPUs and 32 GB ram, to provide a website service. Medical staff and caregivers can upload the images of pressure ulcers and input clinical information from their cellphones or laptops to obtain the results of the automatic diagnosis in Fig 8.

## Discussion

DL relies on large, labeled, and domain-specific datasets to extract features. Determining how to label images of wounds in a manner that achieves diagnostic accuracy is the key. The most challenging task is to label the various tissues of the ulceration. Aside from the boundary hand drawing method, the painting method is commonly used. But the problem remains unsolved; there are no pre-defined regions or boundaries. The inter-rater variability and even the intra-rater variability are still considerable [36]. Algorithms to define meaningful wound areas, such as superpixel segmentation, are required before labeling.

The concept of superpixel segmentation was first proposed by Ren at al [37] to group adjacent pixels with similar features, such as intensities, colors and textures. After processing by algorithms, an image is segmented into many small regions called superpixels. In the last decade, different superpixel algorithms have been proposed, such as Felzenszwalb's efficient graph in 2004, Quickshift in 2008, and simple linear iterative clustering (SLIC) in 2012. SLIC,

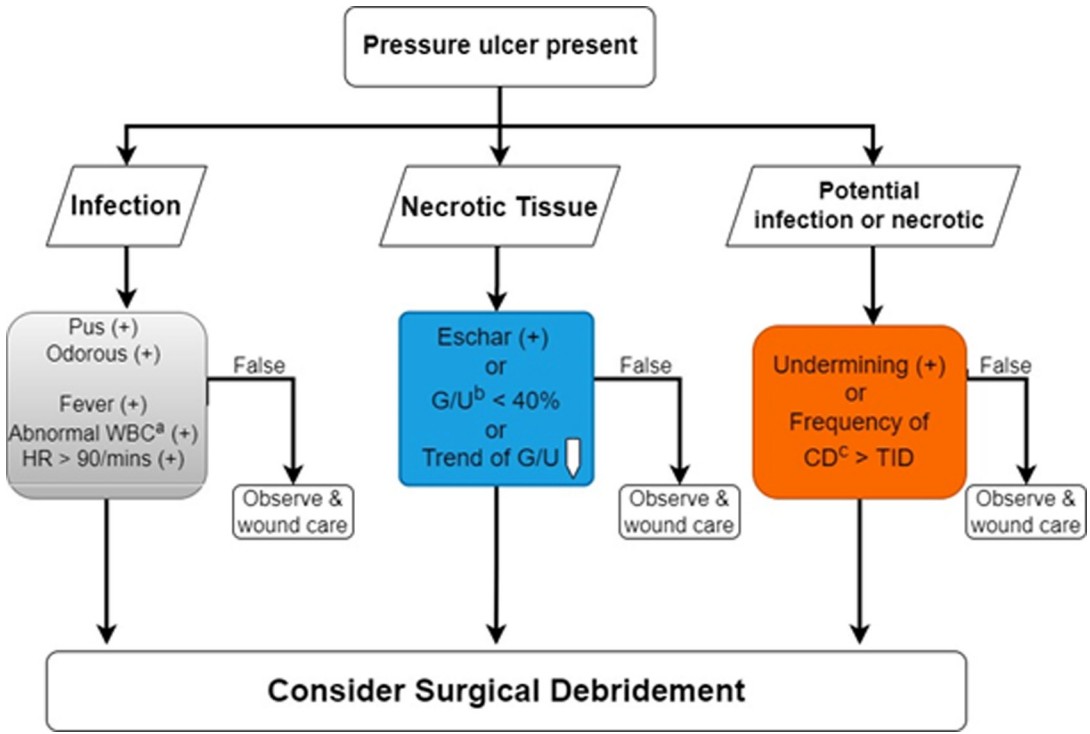

**Fig 7. Decision tree of surgical debridement.** Combining the DL segmentation result (blue) with clinical data (grey) and other wound conditions (grey and orange). [a]Abnormal WBC": white blood cell count> > 12,000 / mm^3. [b]G/U: Granulation / Ulceration Area. [c]CD: Change dressing.

a k-means clustering algorithm, has become the most popular superpixel algorithm [29]. Several modified versions of SLIC have been proposed. For example, linear spectral clustering (LSC) uses a ten-dimensional space to get a better boundary [38]. The superpixel based edge detection algorithm (SBED) is a centroid updating approach to decrease the effect from noise [39]. These versions all demand a higher level of computer calculation. We have adopted the original SLIC to draw boundaries for tissues, because it has three crucial properties: it is fast and simple [40]; it uses a regular shape and similar size; it delivers good adhesion to the object border.

To perform superpixel segmentation before labeling, we reduced the images to 1000*750 pixels. The total number of labeled pixels of each class in the superpixel datasets was far less than the number of pixels labeled with *labelme* (S6 Fig). Despite having less data to work with, the model performs better on the testing images of reduced size (Fig 3). Zhang et al. published an interesting study which helps explain this outcome [41]. When they input randomly labeled objects or random pixels, after 10 thousand steps, neural network models still converged to fit the training set perfectly. The neural networks were rich enough to memorize this highly flawed training data, but their results on testing datasets were poor. This finding indicates that if a model is trained from a dataset with large inter- or intra-rater variability, the segmentation results might be good on the training set but poor on real-world images. Superpixel segmentation improves the inter- and intra-rater by defining the meaningful borders between different tissues.

Before the era of machine learning, radiation oncologists had already faced this challenge. Radiation therapy had been applied to almost every cancer in the human body. A successful treatment relied on a precise hand-drawn tumor contour in a CT or MRI image. The targeted

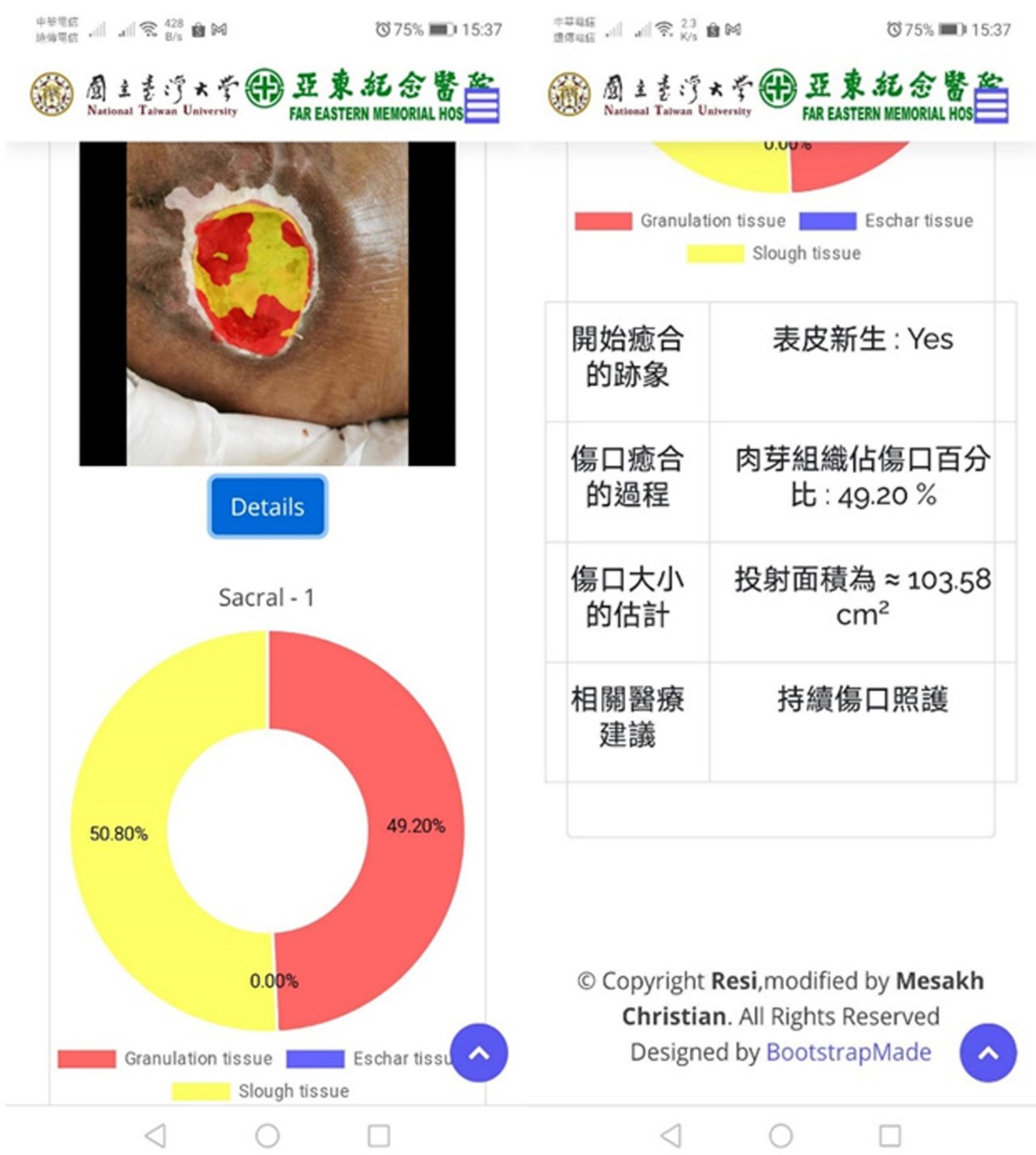

**Fig 8. The print-screen images of our automatic diagnosis server.** Our medical staffs can upload images and clinical data via cellphones.

tumor contour came from the agreement of multiple experts (relative ground truth). However, the inter-rater agreement of such experts ranges from 50 to 90% depending on the tasks and image modalities [42]. To increase agreement, algorithms or models are used to define meaningful regions for medical experts. Radiation experts need only to fine-tune computer-generated borders rather than to hand-draw them on their own. The models are usually trained from a small, but highly reliable labeled dataset consisting of textbook-like medical images

[43]. Otherwise, algorithms are needed that can properly segment images related to ground truth, such as SLIC [44].

It is obvious that increasing the number of superpixels (K) can segment images more accurately (boundary recall) [45, 46]. However, when superpixel segmentation is combined with other ML models, increasing K is not always positively correlated with performance [47]. If the number is too high, the texture information of each superpixel diminishes. In our datasets, the average wound area is around 41.8% of the entire image, and we resized all images to 1000*750 pixels. Our experienced plastic surgeons could easily label all types of tissues when K reached 800 to 1000 without compromising the details (Fig 2). As the number of superpixels was increased beyond 1000, the consensus of labeling became as hard to achieve as it was using the boundary labeling method.

As for data imbalance, Re-ep is an important class but has small number of pixels with irregular shape, which makes it difficult to be detected. Among different classes combination, we found that re-ep training with ulceration had better results than re-ep training with other tissues (granulation, eschar, and slough). This finding cannot be easily explained by data numbers. The total pixels of ulceration were more than the sum of the three types of other tissues (S1 Fig). When re-ep was trained with ulceration, it was actually more in the minority, increasing the degree of imbalance, which should theoretically result in worse re-ep classification [48, 49]. Class overlapping is another factor that will deteriorate classification results. Neither can it be explained by class overlapping because re-ep is more likely to overlap with ulceration than with the other tissues [50].

Another possible factor may be the number of classes. However, studies bearing on this issue are limited and reach contradictory conclusions. In a study of ImageNet (a large dataset of images comprising a large number of classes consisting of a small number of individual objects), Abramovich et al. found that including more classes improves the accuracy of model classifiers [51]. This is because sorting objects into more classes is an operation of supervised feature extraction. Other studies have found that merging classes can improve a classifier's accuracy because it reduces the inter-class labeling errors of the dataset [52]. For example, if we sorted Giant pandas as *Ursus* (bear), it would be wrong because they belong to the genus *Ailuropoda*. Whereas, if we merged *Ursus and Ailuropoda* into the family *Ursidae*, Giant pandas could be classed as *Ursidae* correctly. However, the status of our datasets does not fit the circumstances of the above studies completely. First, our dataset has a different composition from ImageNet, with a small number of classes, each consisting of a large number of individual pixels. Secondly, Re-ep is not a class that can be merged into other classes. Re-ep tissue was more likely to be classified as background when training with other tissues because it lies outside of the wounds, is distinct from the other tissues and comprises a smaller number of pixels.

## Limitations

Pressure ulcer tissue is more complex than our labeling suggests. Pressure ulcers may expose ligaments, periosteum or bone cortex. Though some studies have considered more tissue types [53, 54], we used only three primary tissues (granulation, slough, and eschar) for several reasons. More types of tissue increase the difficulty of labeling and augment inter-rater variability. Classifying more tissues may not affect clinical judgement of wound healing. Granulation is the most critical tissue of wound healing to be observed. However, detection of only three types of tissues has limitations if we want the DL model to output more advanced suggestions, such as the need for reconstructive surgery.

The estimation of wound size is a projection of the 3-dimensional (3D) wound surface onto the 2-dimensional image. The angle of the camera to the wound bed will affect the result of the

projection area. We suggest that the camera should be parallel to the wound bed to get accurate results. Moreover, our segmentation results did not include the depth information of pressure ulcers. To solve the above problems, 3D wound surface images are needed. Special devices are required to gather more images and information, such as stereo images, multiview by structure light guided or light detection and ranging (LIDAR) guided (time of flight (ToF) array).

## Conclusion

We have proposed systematic labeling methods to create pressure ulcer datasets for deep learning. Superpixel preprocessing, a region-based method, was applied for various tissues inside the ulceration, while a boundary-based method was used for re-epithelialization and ulceration. Several powerful DL models were trained and tested on the two datasets and had promising results on the wound segmentation and tissue classification tasks. Combining the segmentation results and other clinical information, our algorithm can detect the signs of wound healing, monitor the progress of wound healing, estimate the size of a wound and suggest the need for surgical intervention.

## Supporting information

**S1 Fig. The number of pixels of each class.** The pixels of ulceration are slightly more than the sum of pixels of granulation, slough, and eschar.
(TIF)

**S2 Fig. The confusion matrix of model 1.** The matrix showed the results when re-ep was trained with ulceration.
(TIF)

**S3 Fig. The confusion matrix of model 2.** The matrix showed the results when re-ep was trained with granulation, eschar and slough (sum up as others).
(TIF)

**S4 Fig. Collage results of tissue classification inside wounds from the Medetec Wound Database.**
(TIF)

**S5 Fig. The workflow of automatic diagnosis and the outputs.**
(TIF)

**S6 Fig. The bar chart of pixel numbers.** The total number of pixels of different tissues in the *labelme* dataset (orange bar) and superpixel dataset (blue bar).
(TIF)

**S1 Dataset. Region-based label method.** https://doi.org/10.6084/m9.figshare.17206904.v1.
(RAR)

**S2 Dataset. Boundary-based label method.** https://doi.org/10.6084/m9.figshare.17206940.v1.
(RAR)

## Acknowledgments

We thank Shih-Chen Huang, who helped collect the images of pressure ulcers and coordinate with the plastic surgeons in labeling.

## Author Contributions

**Conceptualization:** Che Wei Chang, Dun Hao Chang.

**Data curation:** Dun Hao Chang, Tom J. Liu, Yo Shen Chen.

**Project administration:** Che Wei Chang.

**Resources:** Feipei Lai.

**Software:** Che Wei Chang, Mesakh Christian, Wei Jen Chen.

**Supervision:** Feipei Lai.

**Validation:** Tom J. Liu.

**Writing – original draft:** Che Wei Chang, Mesakh Christian.

**Writing – review & editing:** Che Wei Chang, Feipei Lai.

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
