## [Decision Letter · Decision Letter 0]

23 Nov 2021

PONE-D-21-28897Deep Learning Approach Based on Superpixel Segmentation Assisted Labeling for Automatic Pressure Ulcer DiagnosisPLOS ONE

Dear Dr. Chang,

Thank you for submitting your manuscript to PLOS ONE. After careful consideration, we feel that it has merit but does not fully meet PLOS ONE’s publication criteria as it currently stands. Therefore, we invite you to submit a revised version of the manuscript that addresses the points raised during the review process.

We look forward to receiving your revised manuscript.

Kind regards,

Anwar P.P. Abdul Majeed

Academic Editor

PLOS ONE

Journal Requirements:

2. We note that your paper includes detailed descriptions of individual patients/participants. As per the PLOS ONE policy (http://journals.plos.org/plosone/s/submission-guidelines#loc-human-subjects-research) on papers that include identifying, or potentially identifying, information, the individual(s) or parent(s)/guardian(s) must be informed of the terms of the PLOS open-access (CC-BY) license and provide specific permission for publication of these details under the terms of this license. Please download the Consent Form for Publication in a PLOS Journal (http://journals.plos.org/plosone/s/file?id=8ce6/plos-consent-form-english.pdf). The signed consent form should not be submitted with the manuscript, but should be securely filed in the individual's case notes. Please amend the methods section and ethics statement of the manuscript to explicitly state that the patient/participant has provided consent for publication: “The individual in this manuscript has given written informed consent (as outlined in PLOS consent form) to publish these case details

"This work was supported by the Innovation Project of Far Eastern Memorial Hospital (Grant No. PI20200002). We thank Shih-Chen Huang, who helped coordinate with the plastic surgeons in labeling the images of pressure ulcers"

"This work was supported by the Innovation Project of Far Eastern Memorial Hospital (Grant No. PI20200002)"

4. We note that you have stated that you will provide repository information for your data at acceptance. Should your manuscript be accepted for publication, we will hold it until you provide the relevant accession numbers or DOIs necessary to access your data. If you wish to make changes to your Data Availability statement, please describe these changes in your cover letter and we will update your Data Availability statement to reflect the information you provide

Reviewers' comments:

Reviewer's Responses to Questions

**Comments to the Author**

1. Is the manuscript technically sound, and do the data support the conclusions?

Reviewer #1: Yes

Reviewer #2: Yes

2. Has the statistical analysis been performed appropriately and rigorously? 

Reviewer #1: Yes

Reviewer #2: Yes

3. Have the authors made all data underlying the findings in their manuscript fully available?

Reviewer #1: Yes

Reviewer #2: Yes

4. Is the manuscript presented in an intelligible fashion and written in standard English?

Reviewer #1: Yes

Reviewer #2: Yes

5. Review Comments to the Author

Reviewer #1: The authors proposed a workflow of labeling images of pressure ulcers as a way to use them as data for a deep learning algorithm to learn and perform detection and classification of pressure ulcers. The work presented is technically sound and should be useful to others interested in the field.

Good introduction to the background and previous related work. While the sharing of their labeling techniques is appreciated, it is not necessarily "suitable for use with any images of pressure ulcers, and not just for our dataset" as claimed.

3.2 Total and breakdown of the data images collected should be mentioned here. Also were all these images taken with the same parameters such as distance from the wound to the camera, camera angle and such. Obviously, the same wound would look bigger and have more pixels if it is nearer to the camera than if it is further away. This should be clarified.

3.3 This section should be more concise. The division between inside and outside ulceration should be explained more directly. I think this can be explained better using either point form format or listing in down in a table.

3.4 What were the 4 models used to test the segmentation ? And why did the author use 4 different models for this ? It is not mentioned in this section. The text in Fig 2 is too small and unreadable.

3.5 Perhaps the surgeons found it difficult to agree on the regions because they were not specifically defined for them ? As in this paper, there was no specific definition on how the different tissues mentioned were defined. Figure 5 is more confusing than it should be. The flow of work should be clearer and any graphics that are not important should be removed.

5. Discussion

- the discussion is not focused and doesnt expand on the findings and instead introduces new concept and results. There is also quite a lot of information that is repeated from previous sections.

Overall formating needs to be rechecked and redone. For example, most paragraphs are not aligned correctly. Same goes for figures (which still have boxes around them) and equations. While the first half of this paper is quite good, the later half seems rushed and does not reach the quality expected after reading the first half. I hope that the authors can present their work (which seems like a lot of work) in a way that is easier to understand and appreciate.

Reviewer #2: The authors explored the use of different DL models for the segmentation and classification of pressure ulcer. They also explored the efficacy different labelling technique, i.e. manual labelme as well as the SLIC. The paper methodological employed in the study is sound, whilst the results were well discussed. The following are my comments to improve the manuscript.

1. Section 3 - Methods - Naturally the evaluation metrics described in 3.1 should be moved at the end of the methods section.

2. Why was the ResNet101 used as the encoder for all models evaluated? Please substantiate the selection through literature, in the even a sensitivity test was not carried out.

6. PLOS authors have the option to publish the peer review history of their article (what does this mean?). If published, this will include your full peer review and any attached files.

Reviewer #1: No

Reviewer #2: No

---

## [Author Response · Author response to Decision Letter 0]

15 Dec 2021

Dear reviewer 1

3.2 Total and breakdown of the data images collected should be mentioned here. Also, were all these images taken with the same parameters such as distance from the wound to the camera, camera angle and such. Obviously, the same wound would look bigger and have more pixels if it is nearer to the camera than if it is further away. This should be clarified.

Answer: Thank you for this critical observation. The images of pressure ulcers were download from our EMR, which were taken by various types of devices, including cellphones, pads and cameras. The conditions of taking pictures (illumination, background, distance, and angle) were also diverse. We had tried using a small number of images being taken under the standard condition to train models as in previous articles. Although these models had good results only in the training dataset, their performance was poor when facing images of other patients. That is unexpected but not surprised. It is important for deep learning models to comprehend information from any condition.

Since the images were downloaded from the EMR of our hospital, we did not know the exact distance and camera angle of every image. However, in our dataset, the average ratio of wound area to the whole picture is around 44%. The distance too far or wound too small were difficult to be labeled and will be excluded by plastic surgeons.

3.3 This section should be more concise. The division between inside and outside ulceration should be explained more directly. I think this can be explained better using either point form format or listing in down in a table.

Answer: We thank the reviewer for your insightful comment. To address this comment, we had condensed the definition and description of different types of tissues into Table 1.

3.4 What were the 4 models used to test the segmentation? And why did the author use 4 different models for this? It is not mentioned in this section. The text in Fig 2 is too small and unreadable.

Answer: We apologize for not providing clarity while discussing the effect of classes combination. This paragraph demonstrated that a model has better sensitivity and DC for re-ep when re-ep training with ulceration than granulation + eschar + slough, although ulceration ≈ granulation + eschar + slough. In aspect of re-ep, the partitions will result four models, but only models with re-ep were used for comparison. Our description was slightly confusing for which we rewrote the paragraph to be clearer.

3.5 Perhaps the surgeons found it difficult to agree on the regions because they were not specifically defined for them? As in this paper, there was no specific definition on how the different tissues mentioned were defined. Figure 5 is more confusing than it should be. The flow of work should be clearer and any graphics that are not important should be removed.

Answer: As your professional suggestions, we condensed the definition and description of tissues into Table 1. They have specific definitions but still hard to be labeled. Most tissues inside and peri-wound have a various range of colors and texture. It is unlikely to take every piece of tissue on the wounds for biopsy to confirm. Images can only be co-labeled by experienced plastic surgeons as “relative” ground truth. In our initial dataset, there were many excluding images because of no consensus on labeling. Even within the labeled images, there were still pixels difficult to be labeled and left undefined. The relation can be described as Ulceration = (Granulation ∩ Eschar ∩ Slough ∩ Undefined).We had revised Figure 5 to be more legible. We separated the labeling & training from automatic diagnosis to make new Figure 4 and S5 Figure.

5. Discussion- the discussion is not focused and does not expand on the findings, and instead introduces new concept and results. There is also quite a lot of information that is repeated from previous sections.

Answer: We appreciate this critique from the reviewer. The first paragraph of discussion was redundant. We had trimmed down to make it more concise. There were two main parts in the discussion. The first part was the expanding of superpixel segmentation from the evolving of this type of ML. After superpixel segmentation processing, we input fewer pixel numbers but achieved better results in a manner of improving inter-rater consistency. In the second part, we tried to explain why different classes combination has an impact on the recall (sensitivity) of a specific class. Re-ep is an important class but has small number of pixels with irregular shape.

6. Overall formatting needs to be rechecked and redone. For example, most paragraphs are not aligned correctly. Same goes for figures (which still have boxes around them) and equations. While the first half of this paper is quite good, the latter half seems rushed and does not reach the quality expected after reading the first half. I hope that the authors can present their work (which seems like a lot of work) in a way that is easier to understand and appreciate.

Answer: We apologize for not following the format of the journal. We are grateful for your general enthusiasm to our work. We had revised all our tables, figures, and supporting materials, including re-drawing some figures to make them legible. The discussion was also revised to be adhesive to the results.

Best Regards and Merry Christmas

Dr. Che Wei, Chang

---

Dear reviewer 2

We apologize for not following the format of the journal. We are grateful for your general enthusiasm to our work.

1. Section 3 - Methods - Naturally the evaluation metrics described in 3.1 should be moved at the end of the methods section.

Answer: We thank the reviewer for this thoughtful comment. We had moved the evaluation metrics before the results of different combination were presented. We apologize for not following the format of this great journal. We also revised all our tables, figures, and supporting materials, including re-drawing some figures to make them clearer.

2. Why was the ResNet101 used as the encoder for all models evaluated? Please substantiate the selection through literature, in the even a sensitivity test was not carried out.

Answer: We are grateful for this insightful comment by the reviewer. ResNets are powerful encoder that can be compatible with any deep learning models and are easy to optimize. Our preliminary studies had tried ResNet50, ResNet101, and ResNet152 with U-Net. The U-Net with ResNet101 had much better performance than with ResNet50. It is interesting that U-Net with ResNet152 only had a marginal advantage than with ResNet101. ResNet152 had more likely to experience overfitting and consumed more computing resource. Based on the previous result, we applied ResNet101 for all deep learning models in our final presentation. We had added descriptions to Methods- Deep learning models regarding the choice of an encoder.

Best Regards and Merry Christmas

Dr. Che Wei, Chang

---

## [Editor Report · Decision Letter 1]

4 Feb 2022

Deep Learning Approach Based on Superpixel Segmentation Assisted Labeling for Automatic Pressure Ulcer Diagnosis

PONE-D-21-28897R1

Dear Dr. Chang,

We’re pleased to inform you that your manuscript has been judged scientifically suitable for publication and will be formally accepted for publication once it meets all outstanding technical requirements.

Kind regards,

Anwar P.P. Abdul Majeed

Academic Editor

PLOS ONE

Additional Editor Comments (optional): NIL

Reviewers' comments: The author(s) have addressed the queries raised by the reviewers

---

## [Editor Report · Acceptance letter]

8 Feb 2022

PONE-D-21-28897R1 

Deep Learning Approach Based on Superpixel Segmentation Assisted Labeling for Automatic Pressure Ulcer Diagnosis 

Dear Dr. Chang:

I'm pleased to inform you that your manuscript has been deemed suitable for publication in PLOS ONE. Congratulations! Your manuscript is now with our production department. 

Kind regards, 

on behalf of

Dr. Anwar P.P. Abdul Majeed 

Academic Editor

PLOS ONE